# Generalizing Successor Features to continuous domains for Multi-task Learning

## Abstract

The deep reinforcement learning (RL) framework has shown great promise to tackle sequential decision-making problems, where the agent learns to behave optimally through interactions with the environment and receiving rewards. The ability of an RL agent to learn different reward functions concurrently has many benefits, such as the decomposition of task rewards and promoting skill reuse. One obstacle for achieving this, is the amount of data required as well as the capacity of the model for solving multiple tasks. In this paper, we consider the problem of continuous control for robot manipulation tasks with an explicit representation that promotes skill reuse while learning multiple tasks with similar reward function. Our approach relies on two key concepts: *successor features* (SF), a value function representation that decouples the dynamics of the environment from the rewards, and an actor-critic framework that incorporates the learned SF representations. We first show how to learn a decomposable representation required by SF. Our proposed methods, is able to learn decoupled state and reward feature representations. We empirically study this approach on non-trivial continuous control problems with compositional structure built into the reward functions of the tasks.

## 1 Introduction

Reinforcement learning (RL) tackles sequential decision making problems by defining optimal behavior through a reward function, where the agent learns how to behave through interacting with the environment and receiving rewards. The ability of RL algorithms to generalize across different, yet related reward functions, has a great potential to realize more data-efficient algorithms with the capability to transfer to new reward functions. In this paper we look at one particular type of generalization, where the reward function itself changes, however the underlying dynamics of the environment remain the same. This setup is flexible enough to allow transfer happen across tasks, by appropriately defining the rewards which induce different task decompositions. This type of task decomposition potentially allows the agent to tackle more complex problems than, would be possible were the tasks modeled as a single task. We are interested in a setup where the agent is exposed to multiple tasks i.e. tasks with different reward functions. In a multi-goal setting, different reward function can simply be the difference in the euclidean distance to different target goal locations. In a multi-task setting, the difference can be intricately designed in the reward function. For instance a reward function that determines walking forward vs walking backward. We argue that these differences in the structure of the reward function are difficult to capture within a goal or context-conditioned RL frameworks (Sodhani et al., 2021).

In the context of robotics, generalization across tasks is crucial. Consider an agent playing ball games with a racket Figure 1. An agent trained to dribble the ball vs hitting the ball, should be able to quickly learn to play squash, as many of the skills such as approaching and hitting the ball are shared in a more complex task of playing squash. From the learners perspective, all these tasks share the same common properties, the ball falls to the ground due to gravity, depending on heavy it is, and it moves with certain velocity when it is hit by the racket. In other words, all these tasks share common dynamics. What changes is the small details in the reward function. For instance the difference between dribbling a ball vs hitting it against the wall, can be the rotation angle of the racket and the amount of force required.
If it was possible to learn a representation that could decouple such discrepancies between the reward functions, i.e. decoupling the task dynamics and task-related dynamics, one could train an

agent that could re-use the learned representation and quickly fine-tune itself to the more task-specific representation and achieve a faster learning. *Successor features* (SF) (Barreto et al., 2017) is one framework that enables such decomposability of representation, explicitly built into the RL formulation. The main goal of this framework is to promote a desired property where instead of being posed as a decoupled representation learning problem, transfer is instead integrated into the RL framework as much as possible, preferably in a way that is almost transparent to the agent. SFs in theory, enable fast transfer between tasks that differ only in their reward function. The advantage of using an SF framework over model-based RL where one learns models of the reward function, is the ability of dynamics representation re-use which is decoupled from the task-specific representation.

Our main contribution is to address the generalization and expensive inference problem in the classical SF frameworks coupled with GPI (Barreto et al., 2017). We show that a simple architecture can provide a solution to this feature learning problem and demonstrate the effectiveness of our method compared to (Barreto et al., 2020) in more challenging continuous state and action setting. The majority of the existing work using SFs, operates under the discrete action setting or under the GPI setting optimizing a set of policies which in practice are hard to apply on real robotics applications. To the best of our knowledge, our method is the first to show the applicability of SFs coupled with an appropriate representation learn-

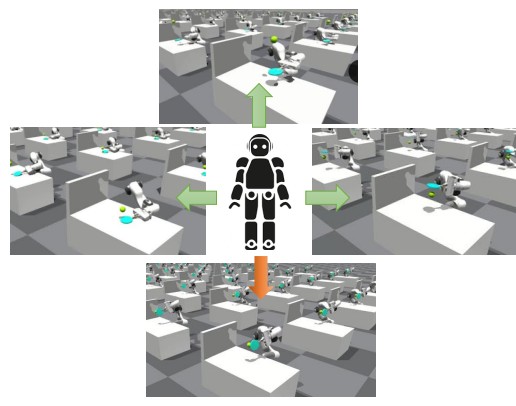

Figure 1: Agent learning various related skills.

ing mechanism solving challenging continuous control tasks. We show that simple modifications to an actor-critic framework can be easily coupled with SFs and empirically demonstrate the efficacy of our method.

To summarize, our contributions are as follows: First we propose a practical implementation of SF framework for continuous state and action domains in the context of actor-critic architecture. Secondly we propose a robust method for learning the representations $\phi$ and $\mathbf{w}$ with the ability to learn disentangled representations for the state-space vs task-specific representation of complex nonlinear reward functions. Finally we demonstrate the efficacy our method on a ranging from tasks including classical continuous control 2D reacher domain on DM control suite (Tassa et al., 2020) to more challenging 3D reacher and manipulation tasks with the Sawyer arm on Meta-World benchmark (Yu et al., 2019).

## 2 BACKGROUND

### 2.1 REINFORCEMENT LEARNING

We assume the interaction between agent and environment can be modeled as a *Markov Decision Process* (MDP (Puterman, 1994)). An MDP is defined as a tuple $M \equiv \langle \mathcal{S}, \mathcal{A}, p, R, \gamma \rangle$ with state space $\mathcal{S}$ and action space $\mathcal{A}$. For each $s \in \mathcal{S}$ and $a \in \mathcal{A}$ the function $p(.|s,a)$ gives the next-state distribution upon taking action $a$ in state $s$, where $p(.|s,a)$ is referred to as the *dynamics* of the MDP. The random variable $R(s,a,s')$ determines the reward received in the transition $s \xrightarrow{a} s'$. Usually we are interested in the expected value of this variable, which is denoted by $r(s,a,s')$, and $\gamma \in [0,1)$ weighs the importance of future rewards. The agent's goal is to find a policy $\pi : \mathcal{S} \to \mathcal{A}$, that is, a mapping from states to actions, that maximizes the value of every state-action pairs, defined as

$$Q^\pi(s,a) \equiv \mathbb{E}^\pi \Big[ \sum_{i=0}^{\infty} \gamma^i r(S_{t+i}, A_{t+i}, S_{t+i+1}) | S_t = s, A_t = a \Big]. \tag{1}$$

where $S_t$ and $A_t$ are random variables indicating the state occupied and the action selected by the agent at time step $t$ and $\mathbb{E}^\pi[.]$ denotes expectation over the trajectories induced by $\pi$. The function $Q^\pi(s,a)$ is referred to as the "action-value function" of policy $\pi$.

RL algorithms based on dynamic programming build on two fundamental operations, *policy evaluation* which is the computation of $Q^\pi(s, a)$, the value function of policy $\pi$ on task with reward $r$, and *policy improvement theorem* (Bellman, 1957). Once a policy $\pi$ has been evaluated, we can compute a *greedy* policy $\pi'(s) \in \arg\max_a Q^\pi(s, a)$ that is guaranteed to perform at least as well as $\pi$, that is: $Q^{\pi'}(s, a) \geq Q^\pi(s, a)$ for any $(s, a) \in \mathcal{S} \times \mathcal{A}$. The computation of $\pi'$ is referred to as *policy improvement*.

## 2.2 Multi-task RL & Transfer learning

In practical situations, agents often face multiple related tasks, such as the robot learning numerous skills in Figure 1. We define tasks $M_i$ drawn from the set $\mathcal{M}$. Then, the goal of multi-task learning is to find $\pi_i^*$, an optimal policy for each MDP $M_i$ with corresponding optimal value function $Q_i^{\pi_i^*}$. Barreto et al. (2017) extended the policy improvement theorem to the scenario where the new policy is computed based on the value functions of a *set* of policies and referred to this as *generalized policy improvement (GPI)*. Suppose the agent has computed $n$ policies corresponding to $Q^{\pi_1}, Q^{\pi_2}, ..., Q^{\pi_n}$ action-value functions. Let $Q_{max} = \max_i Q^{\pi_i}$ and define $\pi(s) \leftarrow \arg\max_a Q^{max}(s, a)$ for all $s \in \mathcal{S}$, then $Q^\pi(s, a) \geq Q^{max}(s, a)$ for all $(s, a) \in \mathcal{S} \times \mathcal{A}$. The only caveat is that it is a waste of computation to compute the value functions of $\pi_1^*, \pi_2^*, ..., \pi_n^*$. This approach becomes appealing if we have a way to quickly compute the value functions of the policies $\pi_i$ on the task $M_{n+1}$.

# 3 Actor-Critic Successor Features

This section will describe Successor Features and their previous use in the discrete action setting. We will then explain our extension, through Universal Value Function Approximators and an Actor-Critic approach, to learn useful Successor Features and corresponding policies for high dimensional multi-task continuous control.

## 3.1 Successor Features Decomposition

Barreto et al. (2017) proposed a simple reward model which leads to the generalization of successor representation (SR) proposed in (Dayan, 1993). The key assumption is that the reward function can be approximately represented as a linear combination of learned features $\phi(s)$. The successor representation (SR) (Dayan, 1993) is a representation that generalizes between states using similarity between their successors, that is, the states that follow the current state given the agent's policy. The generalization of SR with function approximation is referred to as Successor Features (SF) (Barreto et al., 2017) of $(s, a)$ under policy $\pi$. Following (Barreto et al., 2017; 2018; 2020), let $\phi : \mathcal{S} \times \mathcal{A} \times \mathcal{S} \to \mathbb{R}^d$ be an arbitrary function whose output we will see as "features". We assume that there exist features such that the reward function can be written as

$$r(s, a, s') = \phi(s, a, s')^T \mathbf{w} \tag{2}$$

where $\phi(s, a, s') \in \mathbb{R}^d$ are features of $(s, a, s')$ and $\mathbf{w} \in \mathbb{R}^d$ are weights. Intuitively we can think of $\phi(s, a, s')$ as salient events that may be desirable or undesirable to the agent. Based on Eq.2 we can define an environment $M^\phi(\mathcal{S}, \mathcal{A}, p, \gamma)$ as

$$M^\phi \equiv \{M(\mathcal{S}, \mathcal{A}, p, r, \gamma) | r(s, a, s') = \phi(s, a, s')^T \mathbf{w}\}, \tag{3}$$

that is, $M^\phi$ is the set of MDPs induced by $\phi$ through all possible instantiations of $\mathbf{w}$. SFs make it possible to compute the value of a policy $\pi$ on any task $M_i \in M^\phi$ by simply plugging in the representation vector $\mathbf{w}_i$ defining the task. Specifically, if we substitute Eq.2 in the definition of action-value function of a policy we have

$$
\begin{aligned}
Q^\pi(s, a) &= \mathbb{E}^\pi \Big[ r_{t+1} + \gamma r_{t+2} + ... | S_t = s, A_t = a \Big] \\
&= \mathbb{E}^\pi \Big[ \phi_{t+1}^T \mathbf{w} + \phi_{t+2}^T \mathbf{w} ... | S_t = s, A_t = a \Big] \\
&= \mathbb{E}^\pi \Big[ \sum_{i=t}^\infty \gamma^{i-t} \phi_{i+1} | S_t = s, A_t = a \Big]^T \mathbf{w} \\
&= \psi^\pi(s, a)^T \mathbf{w}
\end{aligned}
\tag{4}
$$

One benefit of doing so is that if we replace $\mathbf{w}_i$ with $\mathbf{w}_j$ in Eq.4, we immediately obtain the evaluation of $\pi$ on task $M_j$. This way only relevant module must be relearned, when either the dynamics or reward changes. The key insight of SFs is that linearity of rewards $r_\mathbf{w}$ with respect to the features $\phi$ which gives us the decomposition of the action value of policy $\pi$ on task $r_\mathbf{w}$. In the GPI setting when the agent is presented with a new task $M_{n+1}$, it needs to compute $\{Q_{n+1}^{\pi_1^*}, Q_{n+1}^{\pi_2^*}, ..., Q_{n+1}^{\pi_n^*}\}$, that is, the evaluation of each $\pi_i^*$ under the new reward function induced by $\mathbf{w}_{n+1}$. This in turn would require applying the GPI theorem to the newly-computed set of value functions that will give rise to a policy that performs at least as well as the policy based on any subset of these. Hence (Barreto et al., 2017) proposed to incorporate SFs where the reward function changes to $r_{n+1}(s, a, s') = \phi(s, a, s')^T \mathbf{w}_{n+1}$, as long as we have the correct $\mathbf{w}_{n+1}$ we can compute the value function of $\pi_i^*$ by simply computing $Q_{n+1}^{\pi_i^*}(s, a) = \psi^{\pi_i^*}(s, a)^T \mathbf{w}_{n+1}$. This reduces the computation of all $Q_{n+1}^{\pi_i^*}$ to the simpler supervised problem of approximating $\mathbf{w}_{n+1}$ (Barreto et al., 2020).

Although at first glance, GPI and SFs seem tangled, this characterization of SF does not depend on GPI framework itself. Thus, SF can be used in any RL framework where such decomposition of reward is viable. Furthermore the combination of SFs and GPI provides an elegant framework for transfer in a multi-task setting. However the computational cost of optimizing policies per task is prohibitive in more complex multi-task setting where in the size of task set $M^\phi$ is large.

## 3.2 UNIVERSAL SUCCESSOR FEATURES AND TASK INFERENCE

We start our discussion by assuming the existence of appropriate representations $\phi$ and $\mathbf{w}$ that are a decomposition of the reward function. We then turn our attention to the policy learning phase that will make use of these representations as a substitute for the reward signal. For an RL agent to generalise to unseen tasks, the agent needs to be able to identify and exploit some common structure underlying the task. Two possible sources of structure in this scenario are: i) similarity in the space of tasks, i.e. reward functions, and ii) the shared dynamics of the environment. One framework that enables exploiting structures are *universal value function approximators* (UVFAs) (Schaul et al., 2015). UVFAs extend the notion of value functions to also include the description of a task, thus directly exploiting the common structure in the associated optimal value functions. As discussed in (Borsa et al., 2018), UVFAs and SF & GPI address the transfer problem in quite different ways. With UVFAs, one trains an approximator $\tilde{Q}(s, a, \mathbf{w})$ by solving the training tasks $\mathbf{w} \in \mathcal{M}^\phi$ using any RL algorithm of choice. Borsa et al. (2018) proposed *universal successor features approximators* (USFAs) with GPI. They show that by combining USFAs and GPI can outperform SF&GPI on a grid-world navigation problem. Since majority of existing work explores SFs in the context of GPI or Q-learning under a discrete action setting, in this paper we will examine a more practical UVFA style framework targeted at solving more challenging continuous action control problems.

In this work, we ask the following questions: **Can we incorporate SFs and train a policy which can yield better generalization across multiple tasks? Do SFs introduce implicit composability at the policy level?** Composability of controllers is specially important for real-world applications, where reuse of past experience can greatly improve sample efficiency for tasks that can naturally be decomposed into simpler sub-problems. For instance, a policy for pick-and-place task can be decomposed into (1) reaching specific target, (2) grabbing an object, (3) avoiding certain obstacles. Such decomposable policies can be learned offline without the need to interact with the environment. Although we are not using an explicit notion of compositionality as in (Haarnoja et al., 2018a), we count on such properties emerging from the representation itself.

### 3.2.1 ALGORITHM DERIVATION

In this work we will be extending Soft Actor-Critic (Haarnoja et al., 2018b), by two means of generalisation. That is by combining, UVFA and USFA to learn to generalize values over multiple tasks. The structure of the RL problem creates a number of challenges. Without having access to the target function, we generally use the approximation itself to build targets and many of the techniques like replay buffer and target networks and optimizing two target networks by selecting the minimum which leads to lower variance, are just strategies to remedy this instability (Mnih et al., 2015; Fujimoto et al., 2018). Unfortunately since we are dealing with a multi-dimensional $\psi$ target functions, these instabilities are exacerbated. We elude this problem by taking the average of $\psi$ along the dimension of the features and the minimum is determined based on this average value.

This causes further instability during the policy learning, and longer delayed critic target updates were required. Similar to Hunt et al. (2018), we define action-dependent SF to include the entropy of the policy

$$\psi^\pi(s_t, a_t) = \phi_t + \mathbb{E}_\pi\left[\sum_{\tau=i+1}^{\infty} \gamma^{\tau-t}\big(\phi_\tau + \alpha H[\pi(.|s)]\big)\right] \tag{5}$$

The max-entropy action-value of $\pi$ for any convex combination of rewards $\mathbf{w}$ is then given by $Q_\mathbf{w}^\pi(s, a) = \psi^\pi(s, a)\mathbf{w}$. For more details see Appendix A.1

### 3.2.2   LEARNING SUCCESSOR FEATURES IN CONTINUOUS CONTROL

We now describe how to build a decomposed representation required for $\phi$ and $\mathbf{w}$. To make our discussion more concrete, consider a simple 2D goal reaching environment. We formulate the problem in Eq. 2 as computing an approximate $\tilde{\phi}$ as a multi-task problem, solving the following approximation:

$$\tilde{\phi}(s, a, s')^T \mathbf{w} \approx r_i(s, a, s'), \text{for i} = 1, 2, ..., D \tag{6}$$

We can start our assumption by considering the scenario where such representations $\phi \in \mathbb{R}^n$ and $\mathbf{w}$ exists that satisfy Eq. 2 exactly. For instance, we can define $\phi_i$ as an indicator function signaling important event in the state features e.g. whether an object of type $i$ has been picked up by the agent (Barreto et al., 2020). Analogously, in a continuous control setting, we can consider a goal reaching task with a non-linear reward function. A common reward function for a simple reach task can be defined as the euclidean distance between two vectors, that is, the position of the agent's link and the goal location. Given the state vector $\boldsymbol{\mu} = (x, y)$ and goal vector $\mathbf{g} = (x, y)$, we define the following reward function,

$$r(\mu, g) = -||\mu - g||^2 = ||\mu||^2 + 2g^T\mu - ||g||^2 \tag{7}$$

It is trivial to see the corresponding $\phi(\mu)$ and $\mathbf{w}(g)$ can be recovered as follows,

$$\phi(\mu) = \begin{pmatrix} 1 \\ \mu \\ ||\mu||^2 \end{pmatrix}, \mathbf{w}(g) = \begin{pmatrix} -||g||^2 \\ 2g \\ -1 \end{pmatrix} \tag{8}$$

Thus given such decomposition, we can recover the reward function as the linear combination of $\phi(\mu)^T\mathbf{w}(g)$. Such representation, for this particular reacher task, immediately gives a solution to compute $\psi^\pi$. And by changing the target goal location associated with each reacher task, one can create different tasks, where the only part of the representation i.e. goal location, changing is $\mathbf{w}$, relying on the assumption that the representation captures any task-related features.

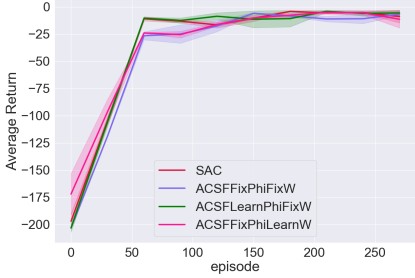

Figure 2: 1-Goal Reacher Task. Comparison between learning a Successor Feature based SAC algorithm with pre-defined $\phi$ and $\mathbf{w}$ vs regressing either $\phi$ and $\mathbf{w}$. These results are sanity checks that given appropriate representations $\phi$ and $\mathbf{w}$, it is possible to incorporate SF within our Actor-Critic continuous action framework.

Now that we have defined $\phi$, we turn to the question of how to determine an appropriate policy $\pi$ to solve the task. We will start with what is perhaps the simplest case, for a single-goal reacher task. In our experiments, Figure 2, we compare the performance of SF integrated with SAC. As a sanity check, we alternate between keeping either $\phi$ or $\mathbf{w}$ fixed and regressing the other. This experiment ensures that our objective is able to learn the decomposed representation for this reward function for each component individually. These results confirm that, given the appropriate successor features, our policy learning mechanism should be able to solve the task, and perform as close to the optimal policy learned via the original reward signal. However most RL tasks have complex reward functions and it is not feasible to always hand-engineer these features. Instead, we are interested in learning these as a pre-training stage.

Barreto et al. (2020) proposed an approach to learn such representation. Given a task $r$, we are looking for $\mathbf{w} \in \mathbb{R}^d$ that leads to good performance of the generalized policy $\pi_\psi(s; \mathbf{w})$ . Suppose we have a set of $m$ sample transitions from a given task, $\{(s_i, a_i, r'_i, s'_i)\}_{i=1}^m$. Then based on Eq.2, we can infer $\mathbf{w}$ by solving the following minimization:

$$\min_{\tilde{\mathbf{w}}} = \sum_{i=1}^m |\phi(s_i, a_i, s'_i)^T \tilde{\mathbf{w}} - r'_i|^p, \tag{9}$$

where $p \geq 1$ and we may want to consider the inclusion of a regularization term in this objective function. If we handcraft $\phi$, we can obtain a solution $\tilde{w}$, which can be plugged in during the representation learning phase. It turns out that Eq.9 can be generalized to also allow $\phi$ to be inferred from data. Given sample transitions from $k$ tasks, $\{(s_{ij}, a_{ij}, r'_{ij}, s'_{ij})\}_{i=1}^{m_j}$, with $j = 1, 2, ..., k$, we can formulate the problem as the search for a function $\tilde{\phi} : \mathcal{S} \times \mathcal{A} \times \mathcal{S} \mapsto \mathbb{R}^c$ and $k$ vectors $\tilde{\mathbf{w}}_i \in \mathbb{R}^c$ satisfying,

$$\min_{\tilde{\phi}} \sum_{j=1}^k \min_{\tilde{w}_j} \sum_{i=1}^{m_j} |\phi(s_{ij}, a_{ij}, s'_{ij})^T \tilde{\mathbf{w}}_j - r'_{ij}|^p \tag{10}$$

where $p \geq 1$. Note that the features $\tilde{\phi}$ can be arbitrary nonlinear functions of their inputs. As discussed in (Barreto et al., 2020) this objective can be decomposed into single objective for learning $\phi$ and $\mathbf{w}$ stage-wise where the initial learned representation $\phi$ can be re-plugged in to learn $\mathbf{w}$. More generally, the problem in Eq.10 can be solved as a multi-task regression (Caruana, 1997).

Barreto et al. (2020) proposed the following procedure for learning $\tilde{\phi}$ and $\tilde{\mathbf{w}}$. First, a random policy $\pi$ is used to collect data from tasks $\mathbf{w}_1, \mathbf{w}_2, ..., \mathbf{w}_k$. They propose to use a neural network with the number of output units or "heads", corresponding to the number of tasks. The top layer of this joint network represents $\phi$, and hence each output unit represents $r_i = \phi^T \mathbf{w}_i$. Given a sample transition $(s, a, r_i, s')$, only the $i$-th output unit of the network has access to the branch where the sample is used to modify $\mathbf{w}_i$ and $\phi$. Once this optimization is done, the weights $\mathbf{w}_i$ are discarded and the same network is used to re-compute $\mathbf{w}$, while keeping the learned representation $\tilde{\phi}$ from previous iteration frozen. This process repeats the optimization process solving for equation 9, but this time solving for $\mathbf{w}$. When learning $\mathbf{w}$, "fresh" output units are added to the network i.e. the head branches of the network weights are randomly initialized again and using gradient descent the associated weights $\mathbf{w}$ are modified, while keeping $\phi$ fixed. Having computed $\phi$ and $\mathbf{w}$, they resume to learning the policy in their GPI setting. (Barreto et al., 2020) demonstrated the effectiveness of this procedure using a simple tabular grid world navigation task. In this paper, we show that this procedure does not allow for learning more complex representation such as a simple reacher task with a euclidean distance reward function. Instead, we propose the following procedure which deemed more appropriate for more complex control tasks with continuous state and action spaces.

### 3.2.3 JOINT TASK INFERENCE

Following (Barreto et al., 2020), we propose optimizing an approximation to the objective in Eq.10 where the representations $\phi$ and $\mathbf{w}$ are learned jointly, as opposed to the two-stage optimization procedure proposed in (Barreto et al., 2020). Given a set of transitions $\{(s_i, a_i, r'_i, s'_i)\}_{i=1}^m$, ideally covering the space of state space for all the tasks seen at training time, we use two neural networks $\phi_\theta$ and $\mathbf{w}_\theta$, where the function $\phi$ takes as input the tuple $(s, a, s')$ and the function $\mathbf{w}$ is a function of the task itself, in a multi-goal setting, this can be the position of the target goal or task ids. Given these two networks, using stochastic gradient descent we can approximate Eq.10 by optimizing,

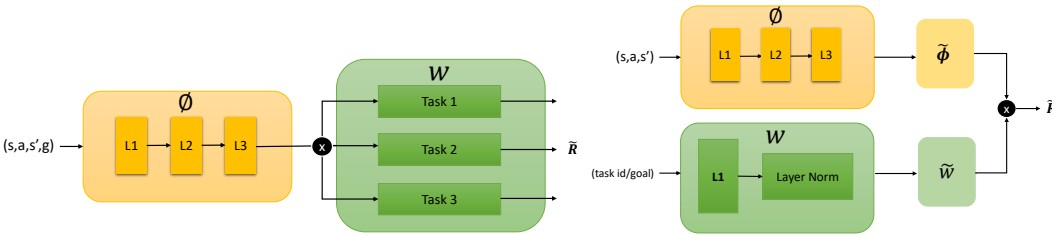

(a) Representation learning in (Barreto et al., 2020).    (b) Our representation Learning Module.

Figure 3: Architecture component of each representation module.

$$\min_{\tilde{\phi}_\theta} \sum_{j=1}^{k} \min_{\tilde{w}_\theta} \sum_{i=1}^{m} |\tilde{\phi}(s_{ij}, a_{ij}, s'_{ij})^T \tilde{\mathbf{w}}(g) - r'_{ij}|^p \tag{11}$$

where $\tilde{\phi}$ and $\mathbf{w}$ are optimized jointly by considering their output as $\tilde{\phi}^T \mathbf{w} \approx r$. This decomposition has two advantages over (Barreto et al., 2020). Firstly, it is easier to decompose the environment dynamics information fed to the network via $\phi(s, a, s')$ and task-specific information $g$ via $\mathbf{w}(g)$ where each of these components are fed the relevant information and the composition of these corresponds to the reward approximation. This is not possible with the architecture proposed in (Barreto et al., 2020) since both $\phi$ and $\mathbf{w}$ share the same input and the decomposition of these representations are made explicit during the stage-wise optimization procedure. In addition, single-stage optimization makes the inference task simpler to train and evaluate. We will illustrate the results of both these approaches in our setup for a simple single-goal and multi-goal reacher task. Figure 3 shows the architecture of the two methods.

## 4 EXPERIMENTS

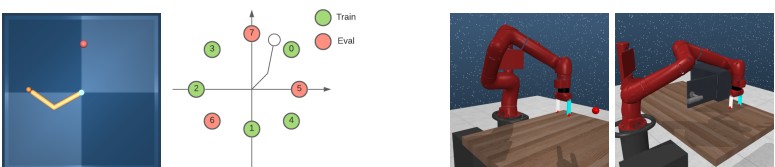

Figure 4: Continuous control tasks we are considering in this work, starting with simple 2D link reacher task to more complex metaworld reacher and door close tasks.

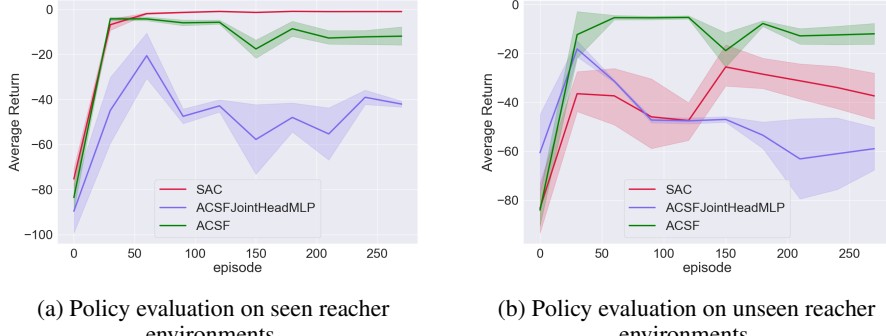

(a) Policy evaluation on seen reacher environments.    (b) Policy evaluation on unseen reacher environments.

Figure 5: Multi-Goal reacher task regression $\phi$ and $\mathbf{w}$ comparison.

Our experiments consist of learning continuous control robot manipulation tasks shown in Figure 4. In the case of 2D link reacher, we fix the goals, separated into a set of training goal targets and test goal targets, and for metaworld, we consider single goal tasks where the tasks are different.

When learning a decomposed representation of the reward function, the function itself should ideally be a dense function. Otherwise it is hard to learn a good representation with a sparse rewards as also pointed out by (Machado et al., 2020)). For this reason, we modify the original reward function for the default 2D Reacher task in DMSuite to a simple L2 norm squared reward function which is easier to learn than a semi-sparse reward function. The original reward function of this task is sparse, where dense reward are given when the agent is in the proximity of the target goal.

For each task, first, we need to learn representations $\tilde{\phi}$ and $\tilde{\mathbf{w}}$. Barreto et al. (2020) proposed to use a random policy to collect the initial data which could be sufficient for a simple grid world navigation task. In our setting, we found that a simple random policy does not allow full coverage of state space. It is possible that a random policy in a small grid-world setting is enough however we found this to be insufficient in a continuous control setting, allowing to learn a generalizable $\tilde{\phi}$ and $\tilde{\mathbf{w}}$. Therefore we used a mixture of expert data and data generated by a random policy to train $\tilde{\phi}$ and $\tilde{\mathbf{w}}$.

Figure 5 shows comparison between the proposed method in (Barreto et al., 2020) and our method where we compare a SAC policy trained with the original reward, our method with *ACSF* where $\phi$ and $\mathbf{w}$ are learned jointly using the training scheme, referred to as *ACSFJointHeadMLP*, proposed in (Barreto et al., 2020) vs our method for learning the joint representation. The task for this experiment is a multi-goal reacher where for the architecture in (Barreto et al., 2020), each goal i.e. task is represented by a separate head of the network. We can see that training a joint architecture becomes harder for two reasons. Since the joint network takes in the input $(s, a, s', g)$, it is not possible to disentangle the inputs to the module learning $\phi$ and the module $\mathbf{w}$. As expected, the performance drops even further, once we remove the goal information from the observations tuple. We prefer to disentangle these observations for the dynamics learning vs any task-related part of the observation. In addition, the training methodology in (Barreto et al., 2020) is a 3-stage training for the representation learning component, meanwhile one must ensure overfitting is not happening at each stage of learning, and this is hard to evaluate in general without having access to any prior ground-truth. Although this is not a direct comparison of the GPI method in (Barreto et al., 2020), the main objective of these experiment is to evaluate the representation learning procedures embedded into our policy learning framework.

Figure 6 shows the generalization capability of our method evaluated on seen goal locations at train time vs unseen goal locations. Finally Figure 7 shows the performance of our method compared to a goal-conditioned SAC policy. Figure 7a shows the performance of both methods on the reacher task where the goal location is perturbed but this is still a single goal task. Figure 7b shows the performance on door close task and finally Figure 7c shows when these tasks are learned simultaneously. These results confirm that our method works in par with a goal-conditioned policy and it has learned the correct features which in turn enable learning robust policies.

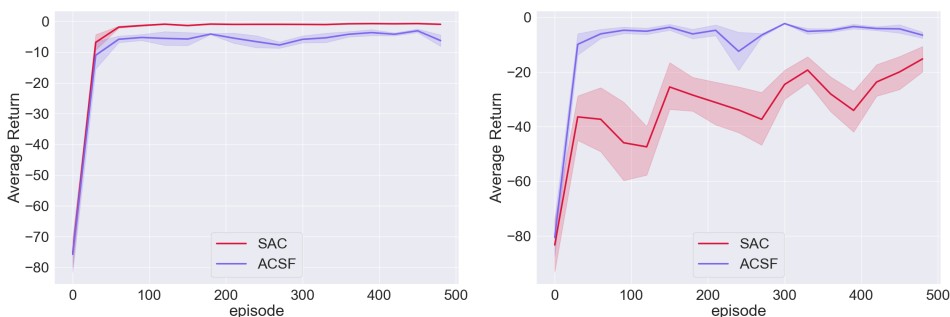

(a) Policy evaluation on seen goal targets.    (b) Policy evaluation on unseen goal targets.

Figure 6: Multi-Goal Reacher Task comparison between SAC and ACSF. This plot shows the training for the train and evaluation setup for the 2Dlink reacher task. The training and evaluation on the same target goals seen at train time show similar performance between a SAC policy and a ACSF policy. However we are interested in the generalization result in Figure 6b where ACSF outperforms SAC on unseen target goals.

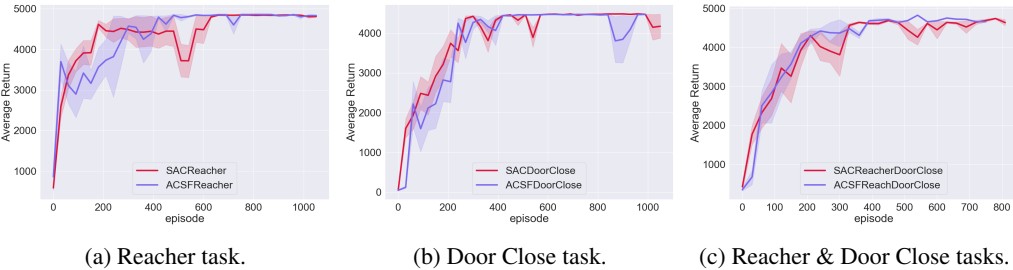

Figure 7: Metaworld reacher task comparison of SAC and ACSF on Reacher and Door close tasks individually, and trained jointly. These results show our method performs in par with learning without an explicit SF representation.

## 5 RELATED WORK

Multi-task RL and representation learning in RL are important topics that have generated a large body of literature and as (Dabney et al., 2021) formally points out, there remains challenging aspect of RL when learning representations. There exist various techniques that inject tasks information directly into the definition of the value function. UVFAs have been used for zero-shot generalization (Mankowitz et al., 2018) and learning a set of fictitious goals previously encountered by the agent (Andrychowicz et al., 2017). Haarnoja et al. (2018a) proposed learning with multiple objectives where they define a compound task expressed in terms of the sum of individual rewards. More recently, (Sodhani et al., 2021) proposed an approach for a contextual multi-task RL as a way to incorporate task metadata, or contextual information. One other common and successful way to approach the representation learning problem is through the use of auxiliary tasks: additional prediction problems that shape the representation used by the agent (Jaderberg et al., 2017; Bellemare et al., 2019).

Another body of work relates to the use of SFs. Kulkarni et al. (2016) presented a generalization of successor representation in a deep reinforcement learning framework, based on DQN with discrete action space. Similar in the spirit, (Zhang et al., 2017), proposes a deep RL architecture built on DQN to solve a robot navigation task. Lehnert & Littman (2019) draws connections between model-based RL and successor features by analyzing properties of different learned latent state spaces. More related to our work is Ma et al. (2020) which combines goal-conditioning with SFs, where SFs are learned end-to-end using temporal difference learning methods. However in their setup, learning these successor features is done implicitly inside their architecture. In addition, they use the true rewards during policy learning and the additional auxiliary loss for learning $\psi$ demonstrates additional stability over DDPG baselines.

Barreto et al. (2019) showed how to combine SF&GPI with deep learning using the reward functions themselves as features for future tasks. Barreto et al. (2020) introduced the representation learning component of SFs as opposed to using hand-crafted features as done in prior work. Hansen et al. (2019) introduces Variational Intrinsic Successor FeatuRes (VISR) which shows that a behavioral mutual information (BMI) maximization provides a solution for learning the successor feature representation. They evaluate their method as an unsupervised pre-training technique. Hunt et al. (2018) introduced SF in the context of maximum entropy framework, extending GPI theorem to max-ent objective. Blier et al. (2021) formally derive a temporal difference algorithm for successor state and goal-dependant value function learning with function approximation. Zahavy et al. (2021) considers a specific class of policy composition called set improving policies (SIPs). Gimelfarb et al. (2021) proposes risk-aware successor features (RaSF) integrated into a generalized policy improvement framework to maximize entropic utilities. Touati & Ollivier (2021) introduces the forward-backward (FB) representation of the dynamics of an MDP, learning two representations instead of one in the SF framework. Their setting is also more tailored towards a goal-oriented RL problems with discrete action space.

## 6 CONCLUSION

Our results suggest that our method can successfully incorporate SFs representation for learning continuous control policies. Learning decoupled representations has been a longstanding challenge in RL, and this work must continue in continuous control tasks, especially for the multi-task setting with continuous goals. Our work sheds light on possible ways of incorporating SFs and their capability to solve more challenging RL control problems. Unlike the framework proposed in (Barreto et al., 2020), which is explicitly tailored for transfer, this is not yet as straightforward in our actor-critic architecture. Future work in this area includes more stable online feature learning and scaling to more complex reward functions. By demonstrating that ACSF performs well on large continuous tasks, we believe our approach is an important step in the direction of composable representations.

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

# A    APPENDIX

## A.1    IMPLEMENTATION DETAILS

The loss function for learning $\psi$ is:

$$\mathcal{L}_{ACSF}(\psi) = \mathbb{E}_{(s,a,s',d)\sim\mathcal{D}}\left[\frac{1}{2}\left(\psi(s,a) - y(\phi,s',d)\right)^2\right] \tag{12}$$

where the target is given by

$$y(\phi,s',d) = \phi(s,a,s') + \gamma(1-d)(\psi(s,a) - \alpha\,log\,\pi_\theta(\tilde{a}'|s')), \quad \tilde{a}' \sim \pi_\theta(.|s') \tag{13}$$

where $d$ corresponds to the termination criterion. The gradient with respect to the parameter $\theta$ is

$$\nabla_\theta \mathcal{L}_{ACSF}(\psi) = \mathbb{E}_{(s,a,s',d)}\left[\left(\psi(s,a) - y(s,a,s')\nabla_\theta\psi(s,a)\right)\right] \tag{14}$$

which is similar to the gradient used in (Haarnoja et al., 2018b), with the distinction that $\psi$ is a multi-dimensional matrix rather than a vector as in the original definition of $Q$. The dimensionality of $\psi$ is a hyperparameter that needs to be tuned depending on the task observations complexity.

Figure 3 shows the comparison of the setup for our representation learning module compared to (Barreto et al., 2020). First stage requires learning the representations $\phi$ and $\mathbf{w}$. Ideally, the data used to train these modules needs to be diverse enough, covering the state space adequately. We found that learning these representation online tangled with policy learning does not perform well since the representation needs to be able to handle regions where policy behavior is optimal. Therefore we resort to using expert policy data. However we still do require the diversity. More intuitively, this is because at the beginning of the agents learning experience, a random policy is exploring, it is important to have good coverage of the state space that can ultimately lead to the highest performing regions. Therefore we use a mixture of expert to random policy data of a $75\% - 85\%$ of expert and the remaining data generated from a random policy with some action noise. We found this ratio to be important for training robust representations that will later on perform well during the SF policy learning stage.

Table 1: ACSF Hyperparameters

| Parameter | Value |
| --- | --- |
| optimizer | Adam (Kingma & Ba, 2015) |
| learning rate | $1.10^{-4}$ |
| discount ($\gamma$) | 0.99 |
| dimension of hidden layers $\psi$ | 1024 |
| dimension of hidden layers **w** | 64 |
| dimension of hidden layers policy | 1024 |
| dimension of samples per minibatch | 256 |
| nonlinearity | ReLU |
| Actor update rate | 1 |
| $\psi$ update rate | 4 |
| Dimensionality of $\psi$ | 4-28 (Task dependent) |

We also regularize the **w** using the layer weight normalization (Salimans & Kingma, 2016) which is a simple reparameterization of the weight vectors in a neural network that decouples the magnitude of a weight tensor from its direction. For the policy learning, we use an open-source SAC implementation [1], extending this implementation to a ACSF policy.

## A.2 APPROXIMATING THE REWARD FUNCTION

We demonstrated our method on tasks with more reaching-style reward functions which are still learnable with our objective function. However this simple objective fails to capture more difficult tasks or tasks with more complex state representations. For instance, our method fails to learn good representations for some of the tasks in Meta-World such as picking and placing objects or tasks interacting with an object such as pressing a button. We hypothesize that more complex reward functions are too difficult for the current method to automatically decompose.

## A.3 FEW-SHOT TRANSFER

Consider the 3 reaching scenarios presented in Figure 8. In our experiments we found that, an agent that had been trained on the reacher and door close task, struggled to zero-shot transfer to the third reacher task with an obstacle on the way. The performance had higher variance than we expected and this happened with both SAC and ACSF which could entail an underlying issue with the policy learning. Note that we would expect the agent to adapt to the third scenario through fine-tuning. However in our experiments we found that the continual learning the third behavior led to slower learning and, in some cases, reaching sub-optimal behavior due to instability. We found that it is better to train an agent on the 3 tasks simultaneously or an agent trained per task separately. We hypothesise that an agent starting from scratch with random exploration has a better chance at discovering all the skills compared to an agent who has learned to reach the first two scenarios. Transferring to the third case, it needs to unlearn the previous behavior and adapt.

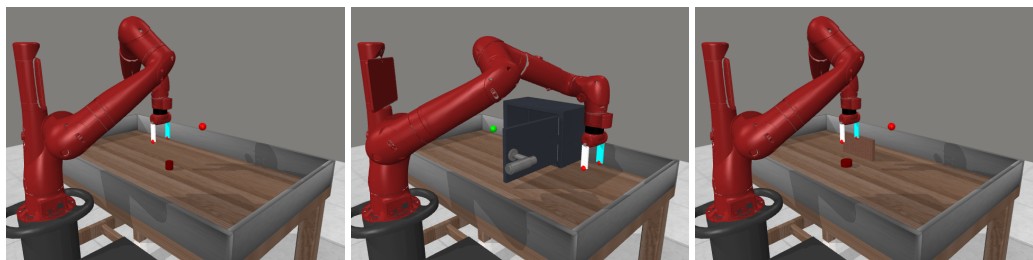

Figure 8: Agent learning various forms of reaching skills.

---

[1]https://github.com/denisyarats/pytorch_sac

---

**Algorithm 1** ACSF- Plugging in learned $\tilde{\phi}$ and $\tilde{\mathbf{w}}$

---

**Input:** initial policy parameters $\theta$
        $\Psi$ parameters corresponding to $\psi_1$ and $\psi_2$
        Replay buffer $\mathcal{D}$
        Set target parameters equal to main parameters
        $\psi_{tar,1} \leftarrow \psi_1,\ \psi_{tar,2} \leftarrow \psi_2$
**for** Time $t = 0$ to $\infty$ **do**
    Observe state $s$ and select action $a \sim \pi_\theta(.|s)$
    Step in environment, $s_t' \sim p(.|s_t, a_t)$
    $\mathcal{D} \leftarrow \mathcal{D} \cup (s_t, a_t, \phi_t, s_t', g)$
    **if** it's time to update **then**
        **for** $j$ in range(numUpdates) **do**
            Randomly sample a batch of transitions, $B = \{(s, a, \phi, s', d, g)\}$ from $\mathcal{D}$
            Compute targets for the D-dimensional $\Psi$ functions:

$$\Psi^D_{\psi_{tar,1}}(s, a, g) = \sum_{d=1}^{D} \Psi_{\psi_{tar,1}}$$

$$\Psi^D_{\psi_{tar,2}}(s, a, g) = \sum_{d=1}^{D} \Psi_{\psi_{tar,2}}$$

$$y(\phi, s', d) = \phi(s, a, s') + \gamma(1 - d)(\min_{i=1,2} \Psi_{\psi_{tar,i}}(s, a, g) - \alpha\, log\, \pi_\theta(\tilde{a}'|s', g)),$$

$$\tilde{a}' \sim \pi_\theta(.|s', g)$$

            Update $\Psi$-functions by one step of gradient descent using

$$\nabla_{\psi_i} \frac{1}{|B|} \sum_{(s',a',\phi,s',d)} \in B(\min_{i=1,2} \Psi(s, \tilde{a}_\theta(s), g) - \alpha\, log\, \pi_\theta(\tilde{a}_\theta(s)|s, g))$$

            Compute the $Q-$function

$$Q1 \leftarrow \Psi^\pi_{tar,1}(s, a, g)^T \mathbf{w}$$

$$Q2 \leftarrow \Psi^\pi_{tar,2}(s, a, g)^T \mathbf{w}$$

            Update policy by one step of gradient ascent using

$$\nabla_\theta \frac{1}{|B|} \sum_{(s,g)\in B} (\min_{i=1,2} Q(s, \tilde{a}_\theta(s), g) - \alpha\, log\, \pi_\theta(\tilde{a}_\theta(s)|s, g)),$$

            where $\tilde{a}_\theta(s)$ is a sample from $\pi_\theta(.|s)$ which is differentiable w.r.t. $\theta$ via the reparameterization trick.
            Update target networks with

$$\psi_{targ,i} \leftarrow \rho\psi_{targ,i} + (1 - \rho)\psi_i$$

---

