# OpenReview forum: "Generalizing Successor Features to continuous domains for Multi-task Learning"
_ICLR.cc/2022/Conference — ICLR 2022 Submitted_

### Official Review · Reviewer_QCxi · 2021-11-02

**Correctness:** 3
**Technical Novelty And Significance:** 2
**Empirical Novelty And Significance:** 2
**Recommendation:** 3
**Confidence:** 4

**Main Review:**

Pros:
- The use of SFs is well motivated in the multi-task learning scenario for continuous domains since they can promote skill reuse and composability. More than that, the proposed method also allows transfer learning among tasks through the task-specific module.

- The experimental evaluation in Figure 6 shows a clear improvement from the proposed method (ACSF) compared to the goal-conditioned SAC policy, which suggests that ACSF is learning meaningful representations that enable generalization across tasks.

- It is interesting the way that ACSF modifies the SAC framework to enable successor features. It requires some detailed design choices to consider double Q-functions and target networks and keep their stability effect.

- The introduced training method (single-stage procedure) is also valuable due to the simplification from the prior method [1].

Major Concerns:

There are two general points of criticism that should be addressed during the rebuttal period. First, the paper proposes a method that is quite similar to prior literature (e.g.,  Ma et al. [2]), which is not contrasted or discussed in the paper. Secondly, the experimental section does not communicate the intentions well: given the motivations for this method, the experiments miss the critical gains of it compared to prior methods. Furthermore, it lacks a solid analysis of the results and limitations. In more details:

- The contribution of this paper is fairly incremental, and the novelty is questionable. Indeed, Ma et al. [2] proposed “Universal Successor Features”  to capture the underlying dynamics of the environment while allowing generalization to unseen goals. It also extends to the continuous scenario, which seems to refute the claim that this paper “is the first to show the applicability of SFs for continuous control tasks”, as mentioned in the Introduction. Furthermore, the paper does not analyze [2] in the Related Work section, despite the similarity. Could the authors discuss this paper from the perspective of the proposed method during the rebuttal session, please?

- Although I think interesting the level of details to enable the SAC framework for successor features, these minor decisions are quite straightforward (e.g., applying average to reduce dimensionality, and replicating the Q-function inference for the double-Q scenario). The inference scheme and training procedure are also inherited from the SAC framework. I understand that they will work simpler and better due to other components of the proposed method, but they do not present novel ideas themselves.

- The evaluation methodology described in Figure 5 trains the compared methods in different MDPs: the SAC Policy using sparse rewards and ACSF using dense rewards. In my perspective, the sparsity should also harm SAC generalization, and, therefore, this comparison seems unfair. I would like to ask the authors if they think that this is an issue and why.

- As described in the Experiments section, the proposed method relies on expert data to work. This is a critical variable to consider (especially in terms of the method’s practicality). In this way, I would say that the use of expert data should be ablated in the proposed method (so that we can evaluate how dependent the method is on this expert data). Another alternative is to also use expert data in the baseline methods (so that we can evaluate how much other methods also take advantage of this expert data).

- The results from Figure 6 seem promising, but they are not analyzed/discussed in the paper. Indeed, the paper does not cite Figure 6 in the text in any circumstance.

- There is no direct comparison between the proposed network architecture and that one from the prior method (Barreto et. al [1]). I understand that the prior method does not extend to the continuous case, but this ablation regarding the network in the proposed SAC framework will help understand how the proposed architecture improves transferability/generalization.

- In section 3.2, the paper states that a central question for its scope is “Do SFs introduce implicit composability at the policy level?”, but I could not find any analysis/evaluation for this. The proposed experiments do not evaluate such property, and we could not conclude whether it emerges from the representation or not.

- The paper lacks some details in the experimental setup (how to collect this expert data, number of random seeds, the dimensionality of  $\psi$ for each task, number of network layers, how the goal is perturbed in the Reacher task, what is exactly the input for the task component in each scenario), making it difficult to reproduce the results. This information should be added to the paper (perhaps through the appendix). I also would like to invite the authors to release the source code for the ACSF implementation and for the experiments if this is possible (not required), which will address all the issues related to reproducibility.

- Figure 7 shows that the proposed method performs on par with a goal-conditioned policy, as described in the text. However, it is not clear what is the intention behind this experiment: if they perform equivalently, what makes the use of successor features a good idea? It would be simpler to use the goal-conditioned baseline if there is no advantage.

Minor Concerns:
- The paper introduces some symbols in the text without any definition (for example, $\phi$, $\psi$, $\boldsymbol{w}$). It makes the text harder to follow in the first read.
- In section 3.2.2, “We forumlate” -> “We formulate”
- In section 7, Fig 7c is cited twice, but the first time should be Fig 7b (door close task)
- In Table 1, the number of layers seems very high (1024, 64, 1024). Perhaps it refers to the dimension of these layers?

Additional Suggestions:
There are additional suggestions. They are not necessary for acceptance, but in my perspective, they could improve the evaluation of the proposed method.

- I recommend introducing other Representation Learning baselines as comparison methods. There are other works with different representation mechanisms [3,4,5] and such comparison will help validate the use of SFs for representation in the continuous multi-task setting.
- Although is implicit that the proposed method improves the inference problem from the prior method [1], a proper evaluation (mathematical or experimental) would help understand it objectively  (i.e., “how much” it improves). The paper claims that this is the main contribution of the method, but this is not highlighted during the evaluation.
- An interesting way to evaluate the final representation is to plot them in a lower-dimensional space. I would suggest plotting these representations (dynamics and task-specific) to evaluate interpretability and disentanglement visually.

References:

[1] André Barreto, Shaobo Hou, Diana Borsa, David Silver, Doina Precup. Fast reinforcement learning with generalized policy updates. Proceedings of the National Academy of Sciences, 2020.

[2] Chen Ma, Dylan R. Ashley, Junfeng Wen, Yoshua Bengio. Universal Successor Features for Transfer Reinforcement Learning. CoRR, abs/2001.04025, 2020. URL http://arxiv.org/abs/2001.04025

[3] Karol Hausman, Jost Tobias Springenberg, Ziyu Wang, Nicolas Heess, Martin Riedmiller. Learning an Embedding Space for Transferable Robot Skills. International Conference on Learning Representations, 2018.

[4] Amy Zhang, Harsh Satija, Joelle Pineau. Decoupling Dynamics and Reward for Transfer Learning. CoRR, abs/1804.10689, 2018. URL http://arxiv.org/abs/1804.10689.

[5] Dibya Ghosh, Abhishek Gupta, Sergey Levine. Learning Actionable Representations with Goal-Conditioned Policies. CoRR, abs/1811.07819, 2018. URL http://arxiv.org/abs/1811.07819.





**Summary Of The Paper:**

The paper proposes a method to incorporate Successor Features (SFs)  in domains with continuous state and action spaces. It proposes an actor-critic architecture (a variation of the Soft Actor-Critic method) that learns disentangled representations for the environment dynamics and the tasks. The network architecture guarantees such disentanglement with two independent modules: one for the representation of the dynamics $\phi$ (fed by the current state, action, and next state) and one for the task representation $\boldsymbol{w}$ (fed by task-specific information, such as the goal). These modules are learned jointly in a single-stage training procedure, contrasting prior work [1]. The main contribution of this model is the enablement of the SFs for continuous domains without relying on the costly inference mechanism from the classic SFs framework implementation while enabling generalization among similar tasks.

**Summary Of The Review:**

Although the paper is well motivated and presents interesting implementation details, the proposed method misses the discussion of concrete similar work. This related work makes the proposed method seem fairly incremental and only marginally novel. Furthermore, the evaluation setup presents some flaws that make it hard to validate the method and its limitations. Therefore, my perspective is that this paper does not meet the requirements for acceptance.

---

> ### Author Response · Authors · 2021-11-18
> **Response to QCxi**
>
> We greatly appreciate your thorough review and have made several adjustments to the paper based on your suggestions. We have addressed concerns raised with regards to the related work in our main response. As for the remaining points you have raised:
>
> > The experiments miss the critical gains of it compared to prior methods.
>
> Majority of prior methods operate under a discrete action setting. And in this work we are tackling continuous control tasks. The closest work relates to the representation learning component in [1] and we do make comparisons with them. Lack of source code in other related work, and operating under custom environments in many of the existing SFs literature makes it more challenging to benchmark various methods. Hence why we hope that our work and source code can encourage more work on more standardized control benchmarks.
>
> > The SAC Policy using sparse rewards and ACSF using dense rewards. This comparison seems unfair.
>
> The SAC policy does not operate under a sparse rewards setting. The main comparison here relates to the representation learning module between our method and our baseline method. The whole argument being that, having this decomposed architecture is helpful in a sense that we can learn a better representation which in turn yields a better performing policy.
>
> > The proposed method relies on expert data to work.
>
> The use of expert data relates more to the representation learning aspect of SF and less to the policy learning. [1] uses data from a random policy to learn the features $\phi$ and $w$. But their task is a simple low-dimension grid-world navigation problem. Our guess is, the state-space is small enough for a random policy enabling learning an adequate representation. Since our setting is more complex, we need to make sure that there’s fairly good coverage of the whole state-space. This becomes tricky because
> the SFs will give you the value function for the reward $r(s,a,s’|g) = \phi(s,a,s’)^T w(g)$ and if this is not aligned with the real reward, the corresponding value function obtained via SFs will not be optimal.
> We need good representations for $\phi$ and $w$ to be able to use SFs, but we also need to generalize since we are substituting our reward signal throughout the training. It can also be viewed as a limitation since the generalization of the representation is tied to having representative data. An end-to-end approach or alternating between representation and policy learning can be interesting to explore, however empirically we found it challenging to train with our setup.
>
> > There is no direct comparison between the proposed network architecture and that one from the prior method [1].
>
> There is some ambiguity in terms of comparing these two methods. First, to make this a fair comparison, we would discretize our tasks action space to be able to apply methods such as Q-learning. In this case, we would be comparing the two representation learning modules embedded into their own policy learning framework. However we felt that the custom grid-world environment proposed in [1] was tailored to their method of representation learning. For instance they define $\phi$ features as an indicator function signaling whether an object of type $i$ has been picked up and $w$ is a vector encoding per task such as $w=[0, 1]$. Such features might already be distinct enough for any architecture to work, however we believe our method and setup is more general and applicable to any multi-goal/multi-task setting.
>
> > ... “Do SFs introduce implicit composability at the policy level?”, but I could not find any analysis/evaluation for this.
>
> This claim is not explicitly analyzed, however given the generalization results of a policy on novel tasks, one would expect such decomposability to have had occured when sharing $\phi$ among tasks.
>
> > Figure 7 shows that the proposed method performs on par with a goal-conditioned policy ...
>
> We believe SFs have more potential compared to goal-conditioned policies. Since majority of existing work in literature, seek ways to condition policies on some embedding (this can be language embedding  or any other information the policy can be conditioned on). We would expect the two methods to perform on-par on the trained tasks, but perform differently on new related tasks.
>
> > Although is implicit that the proposed method improves the inference problem from the prior method [1], a proper evaluation (mathematical or experimental) would help understand it objectively
>
> It is correct that our only comparison of these is in the context of policy learning and observing the final performance.
>
>
> ### References
> [1] Andre Barreto, Shaobo Houa , Diana Borsaa , David Silvera , and Doina Precup. Fast reinforcement learning with generalized policy updates . https://www.pnas.org/content/117/48/30079

---

> > ### Comment · Reviewer_QCxi · 2021-11-30
> > **Final thoughts based on the rebuttal discussion**
> >
> > I appreciate the time and efforts from the authors to incorporate the feedback from the review and also to discuss some concerns.
> >
> > Although I can see improvements in the paper structure and a better understanding of the paper goals, my perspective is that the evaluation setup keeps missing critical points to support the main claims of the paper.
> >
> > - There is some difference from [2] to the proposed paper, but there is no direct empirical comparison, relying the direct comparison upon an intuition. I agree that the proposed method brings a strong inductive bias for decomposability, but it is also hard to objectively verify how effective/meaningful are the representations from the proposed work based on these experiments.
> >
> > - Still in the same line, this difficulty to evaluate the proposed method is partly because of the fact that the baselines and the proposed method are compared in different scenarios. We should not disregard the change in the MDP for the SAC policy or the use of expert data, since we have different research lines in RL that show how important these variables are for exploration and training stabilization. If they are necessary to make the baseline work, perhaps this is not the right baseline to consider.
> >
> > As final summarization, my point of view is that we can analyze the paper from two different perspectives: first, how effective the method is to absorb SFs and improve generalization based on it; second, how these SFs are "meaningful" and can be used for interpretability or downstream tasks.
> >
> > For the first point, I would say that only the experiment from Fig. 6 is partially aligned to this direction, but the baseline is not valid, the results are limited to a single environment, and a more deep ablation is required to conclude effectiveness (since that the difference from goal-conditioned SAC and the proposed method is not only the SFs themselves, but also the operations to enable it to continuous control).
> >
> > For the second point, judging this property of SFs solely by the performance on unseen goals is vague, especially if we take into consideration the lack of transfer raised in Appendix A.3. In this perspective, I reiterate my additional suggestions, as I think they will provide guidance towards this direction.
> >
> > The work has a very interesting motivation, but I think it should be matured more to meet the acceptance bar from ICLR, especially in terms of evaluation methodology. Therefore, I keep my score.

---

### Official Review · Reviewer_f8Lk · 2021-11-03

**Correctness:** 3
**Technical Novelty And Significance:** 2
**Empirical Novelty And Significance:** 2
**Recommendation:** 3
**Confidence:** 4

**Main Review:**

There are a number of concerns about this work.

1. The paper states that it introduces an actor-critic approach to SF in continuous action spaces. The paper outlines the approach to learning the action-value (and successor features which can be used to compute the action-value function) [the critic], but the only details on how the actor (policy) is represented, trained or generalizes is provided implicitly in the appendix. In particular, a key challenge in continuous action spaces is that it is, in general, non-trivial to find the optimal policy even given the action-value function. How to benefit from the SF representation given this challenge is non-obvious (since SF approach provides an action-value function for a new task, but unlike in small discrete action spaces, this does not automatically allow you to infer a good policy). The challenges of doing this in continuous action spaces is outlined in e.g.\ Hunt et al. (2018).

The approach used here appears to be (from the appendix) that a goal condition policy is learned. Therefore, after training successor features are not used (since the policy does not directly use them), and the test performance is simply based on goal conditioned policies. This seems like a significant limitation of this approach, since it means that the structure of successor feature representation is only able to be used during training. It also seems a bit misleading to suggest in the introduction some sort of solution to successor features is provided for continuous action spaces, when it is more just ignored and SFs are not used after training (and therefore the claims of more structured generalization don't apply). Given this limitation some discussion about why SAC does not generalize as well to unseen tasks would also be helpful since it is also learning a goal conditioned policy and its not obvious why it should be inferior.

2. I have some concerns about the novelty of this work. Hunt et al. (2018) applied successor features to the max-ent RL framework. Kulkarni et al. (2016) learned jointly $\phi$ and $w$. Goal conditioned policies are an old idea. I think combining existing ideas can be of interest, if for example, it demonstrates compelling empirical results. However, the empirical results here are not particularly challenging tasks, and there is no comparison with other goal-condition policy learning approaches (such as Hindsight Experience Replay).

It would be helpful if the authors consider releasing code for reproducing the experiments in this paper. Particularly, as there are a number of implementation details that are unclear.

This work does appropriately cite and explain prior work in this topic.

Minor:

Inconsistent vector notation ($\mathbf{w}$ is bolded, by e.g. $\mu = (x, y)$ is not.

**Summary Of The Paper:**

This work looks at the use of successor features for solving simple continuous control tasks (in particular, reaching to different locations and door closing). The two contributions they enumerate are a ``practical implementation of SF framework for continuous state and action domains'' and jointly learning $\phi$ and $w$ (that is the successor features and the task weights $w$). They show their approach outperforms a goal-conditioned SAC baseline on these control tasks including generalizing to target locations that were not used during training.


**Summary Of The Review:**

This paper puts together an approach to apply SFs to some robotic control tasks. However, the approaches are not particularly novel, the empirical results are not compelling and key details regarding training of the actor are not included.

---

> ### Author Response · Authors · 2021-11-18
> **Response to f8Lk**
>
> Thank you for your review. We have made several adjustments to the paper based on your suggestions and would like to discuss some of the finer points below. We have addressed concerns raised with regards to the related work in our main response.
>
> > .. How to benefit from the SF representation given this challenge is non-obvious.
>
> We agree with your point being raised here in terms of the true benefit of SF. However our framework does not intend to tackle the transferability aspect of SFs but rather one method for learning and making use of such features embedded into an AC architecture. Albiet I agree currently SFs usability is very limiting to be applicable to tasks with complex rewards which is mostly the case with environments deep RL algorithms can currently tackle. We hope that our work can slightly bridge this gap.
>
> > The approach used here appears to be that a goal condition policy is learned ... It also seems a bit misleading to suggest in the introduction some sort of solution to successor features is provided for continuous action spaces.
>
> Our claim is merely a method for learning decomposed representation for SF which is built upon the ideas in [1] and one way of integrating these features to a universal value functions (more generally GVFs) [2]
> that tackles continuous action policy learning. We acknowledge the limitation of our work being that it does not promote a seamless transfer of policies since we are learning a universal policy as opposed to a GPI framework.
>
> > Given this limitation some discussion about why SAC does not generalize as well to unseen tasks would also be helpful since it is also learning a goal conditioned policy and its not obvious why it should be inferior.
>
> We believe this issue is stemming from the way we are structuring our multi-task policies (conditioning on a goal variable). There are more suitable architectures tailored at multi-task learning (multihead SAC being the most naive architectural trick) -  The reason we opted-in for a simple goal conditioned policy was to keep the SF framework as simple as possible while maintaining a fair comparison.
>
> ### References
> [1] Andre Barreto, Shaobo Houa , Diana Borsaa , David Silvera , and Doina Precup. Fast reinforcement learning with generalized policy updates . https://www.pnas.org/content/117/48/30079
>
> [2] Schaul, T., Horgan, D., Gregor, K. and Silver, D., 2015, June. Universal value function approximators. In International Conference on Machine Learning (pp. 1312-1320).http://proceedings.mlr.press/v37/schaul15.html

---

> > ### Comment · Reviewer_f8Lk · 2021-11-27
> > **Continuous action policies**
> >
> > I will respond in two comments.
> >
> > Thanks to the authors for responding confirming that my understanding was correct regarding how they deal with continuous action policies. "However, our framework does not intend to tackle the transferability aspect of SFs but rather one method for learning and making use of such features embedded into an AC architecture. [Albeit] I agree currently SFs usability is very limiting to be applicable to tasks with complex rewards which is mostly the case with environments deep RL algorithms can currently tackle. We hope that our work can slightly bridge this gap." My main feedback is that the approach you are using for the actor and limitations should be explained in the paper.
> >
> > "We acknowledge the limitation of our work being that it does not promote a seamless transfer of policies since we are learning a universal policy as opposed to a GPI framework." Again, is this acknowledgement in the paper (its possible it is an I missed it).

---

> ### Comment · Reviewer_f8Lk · 2021-11-27
> **Response to authors**
>
> I think that author's for their clarifying responses and as raised above have suggested that some of the limitations and clarifications discussed in their response be incorporated in to the paper.
>
> I also appreciate the author's clarifying the relationship between this work and previous work. I agree that the approach is not identical to any existing work. However, (in agreement with the other reviewers) I continue to be concerned that the differences to prior work are relatively nuanced and it lacks comparison with any other approaches to SF learning or representation learning and it has significant limitations (such as not allowing for transfer learning). For this reason, I unfortunately have to leave my recommendation unchanged.

---

### Official Review · Reviewer_Nsua · 2021-11-06

**Correctness:** 4
**Technical Novelty And Significance:** 1
**Empirical Novelty And Significance:** 1
**Recommendation:** 3
**Confidence:** 4

**Main Review:**

**Strengths**: The authors proposed a principled method for incorporating successor feature into an actor-critic algorithm and the decomposition of successor features as $\phi(s,a,s)^Tw(g)$ allows it to adapt to new goals as shown through reacher envs.

**Concerns**:
(i) The proposed decomposition $\phi(s,a,s)^Tw(g)$ feels very similar to that used in (Ma et. al, 2020). Moreover, (Ma et. al, 2020) also incorporates the learned successor features into an actor critic framework and uses it for continuous control tasks. Hence, I am not sure what novelty the proposed work is bringing.

(ii) The environments used in the experiments are limited (only reacher and door close tasks are considered). Moreover, the performance of the proposed method is very similar to the goal conditioned SAC baseline on meta world tasks (Figure 7)

(iii) At an intuitive level, if $\phi(s,a,s')$ is capturing the dynamics and $w(g)$ is capturing the reward information, shouldn't $w$ be also conditioned on $s$ as well (i.e. the final decomposition being $\phi(s,a,s)^Tw(s,g)$). This is because $g$ alone doesn't give reward information. Both $s$ and $g$ are needed for reward information.

**References**
1. Universal Successor Features for Transfer Reinforcement Learning. Ma et. al, 2020. https://arxiv.org/pdf/2001.04025.pdf

**Summary Of The Paper:**

The paper proposes (i) a decomposition for successor features $\phi(s,a,s)^Tw(g)$ which allows the Q-value function to adapt to new goals easily and (ii) then incorporate the learned successor feature representation into an actor critic algorithm (SAC) for goal conditioned tasks (or for solving multiple tasks). The proposed method is tested on continuous control tasks.

**Summary Of The Review:**

Weighing the strengths and concerns as listed above, I am currently recommending the paper for a rejection.

---

> ### Author Response · Authors · 2021-11-18
> **Response to Nsua**
>
> Thank you for your review.
>
> > The environments used in the experiments are limited ...
>
> We found training environments with more complex reward functions resulted in learning poor representations. We believe this is due to the linearity assumption we make when optimizing Eq.(10). This is why we think it is important to understand the current limitations of SF frameworks and
> tackle these issues if we want to scale SFs to more complex tasks. This is one simple mechanism for learning these features, but one direction of future work would be better objectives for learning these decomposed representations.
> We believe none of the existing work in the literature related to SFs, do complex tasks beyond grid world navigation, and the most complex task tackled is usually a 2-3 goal reacher task due to the challenges discussed.
>
> > At an intuitive level $\phi(s,a,s')$ is capturing the dynamics and  is capturing the reward information, shouldn't  $w$ also be conditioned on states so $w(s,g)$.
>
> We think this boils down to the reward structure and the task itself. Consider our toy reacher example, if we were to learn such decomposed representation for the reaching reward, then in theory $w$ does not care about the states and only the target goal location. Let's take another example, consider a reward function for hitting a ball from the top (dribble) vs from the side to play squash. In this case the dynamics for both are shared, and what changes could be the angle of hitting the ball. In that  case, $w$ should capture anything related to the angle, and $\phi$ should capture the dynamics.

---

### Author Response · Authors · 2021-11-18
**We thank the reviewers for their time and providing valuable feedback.**

With the advice of the reviewers, we have made modifications to the paper highlighting the following points:

- We have modified the related work section in an attempt to be more precise about the differences between our work compared to [1] and [2].
- We have included the source code to resolve any ambiguity about the implementation. We hope this will encourage more future research towards this direction.

Further, we would like to provide a joint response to the following points with regards to the related work:

We would like to point out the differences between our method to [1]. In their work, when training the value function $Q(s,a,g)$ they use the actual reward as opposed to the disentangled reward $r(s,a,s'|g) = \phi(s,a,s')^T w(g)$. This is highlighted in Eq.(3) and Eq.(4). And their final loss is $L = L_Q + \lambda . L_{\psi}$, where $\lambda$ is a hyperparameter and in some of their experiments this is a small value (see Table 2 and 3 for $\lambda$ weight of $1e{-6}$).
Note that using these two as training signal is very different, the SFs gives the value function for the reward $r(s,a,s'|g) = \phi(s,a,s')^T w(g)$. And in their case there is no explicit learning of this decomposition $\phi(s,a,s')^T w(g)$. Rather this is just implicitly embedded into their architecture which is trained end-to-end. I would argue such training scheme is misleading because there is no way to verify whether the learned SF representations are meaningful. The learned SFs might be ignored, especially in the case with very small $\lambda$.
The experiments seem to show slight improvements with respect to a baseline (DQN/DDPG) but the additional stability observed in the final performance could be due to the additional feature embedding introduced into the network.
We believe that this is not the conventional way of learning SFs, and using the original reward signal slightly defeats the purpose of using SFs in the first place.

Our goal is to highlight the challenges involved in learning these features in depth which is more aligned with the work in [3] and demonstrate their capabilities under more complex continuous control tasks. In addition, the experimental setup proposed in [1] is closer to a multi-goal training since they train a reacher task with randomly perturbed goals as you would in a multi-goal setting. Their setup doesn't evaluate tasks on unseen goals since all the goals are coming from the same distribution whereas we attempt to show this with our reacher setup in Figure(5).

Furthermore, [2] shows that
the GPI theorem [4] holds for maximum
entropy policies which means they are optimizing multiple policies and this indeed does promote transfer since you can pick the best matching action-value function for the new unseen task, and fine-tune $w$ to quickly compute a new policy for this action-value function corresponding to the new task. Lastly, [5] also operates under a discrete action setting however our setup is tailored for continuous action settings.


### References:
[1] Chen Ma, Dylan R. Ashley, Junfeng Wen, Yoshua Bengio. Universal Successor Features for Transfer Reinforcement Learning. CoRR, abs/2001.04025, 2020. URL http://arxiv.org/abs/2001.04025

[2] Jonathan J Hunt, Andre Barreto, Timothy P Lillicrap, Nicolas Heess. Composing Entropic Policies using Divergence Correction. https://arxiv.org/pdf/1812.02216.pdf

[3] Andre Barreto, Shaobo Houa , Diana Borsaa , David Silvera , and Doina Precup. Fast reinforcement learning with generalized policy updates . https://www.pnas.org/content/117/48/30079

[4] André Barreto, Will Dabney, Rémi Munos, Jonathan J. Hunt, Tom Schaul, Hado van Hasselt, David Silver. Successor Features for
Transfer in Reinforcement Learning. https://arxiv.org/pdf/1606.05312.pdf

[5] Tejas D. Kulkarni, Ardavan Saeedi, Simanta Gautam, Samuel J. Gershman. Deep Successor Reinforcement Learning.
https://arxiv.org/abs/1606.02396

---

### Decision · Program_Chairs · 2022-01-20

**Decision:**

Reject

**Comment:**

The paper presents an actor critic type of method consisting of two types of features -- dynamics and tasks, in the multi-task continuous control setting. While the topic of the research is interesting and relevant to ICLR, the reviewers have concerns with the novelty and technical significance of the work. Specifically, the proposed method is very similar to several other works leading to an incremental novelty. In addition, the method is evaluated only on simple environments. The concerns remain after the discussion period.

In the next version of the manuscript, the authors are encouraged to pursue more difficult settings and modify the method to work on those problems. That would make the paper stronger, and lead to a more novel method evaluated on harder problems.